# Changes in medical student attendance and its impact on student educational outcomes: a systematic review protocol

Palaniappan Ganesh Nagappan [ID],[1,2] Samuel Brown [ID],[1] Alex McManus [ID],[1] Sarah Sayers [ID],[1] Shazia Absar [ID],[1,2] Sapphire Rou Xi Tan [ID],[2] Isla Kuhn,[1] Edward Lau [ID],[2] Charlotte Tulinius [ID] [1,2]

[1]School of Clinical Medicine, University of Cambridge, Cambridge, UK
[2]Department of Public Health and Primary Care, University of Cambridge, Cambridge, UK

**Correspondence to**
Dr Palaniappan Ganesh Nagappan; pgn25@cam.ac.uk

## ABSTRACT

**Introduction** The COVID-19 pandemic has had a significant impact on medical education, with many institutions shifting to online learning to ensure the safety of students and staff. However, there has been a decline in in-person attendance at medical schools across the UK and worldwide following the relaxation of social distancing rules and the reinstation of in-person teaching. Importantly, this trend has been observed prior to the pandemic. While reflected within the literature, there is currently no systematic review describing these changes. We aim to find out how medical students' attendance is changing as documented within the literature and its impact on their educational outcomes.

**Methods and analysis** This systematic review will follow the guidelines of the Centre of Research and Dissemination, Meta-analyses of Observational Studies in Epidemiology and Preferred Reporting Items for Systematic Reviews and Meta-Analyses. We will search the major databases of Medline via Ovid, Embase via Ovid, Scopus, Web of Science, British Education Index via EBSCOhost and ERIC via EBSCOhost.

Two reviewers will independently screen each paper and extract data, with a third reviewer for dispute resolution. All studies reporting on medical students from various universities, both graduate and undergraduate and describing changes in attendance and/or students' educational outcomes will be included. Risk of bias in individual studies will be assessed using the Newcastle-Ottawa Scale and confidence in cumulative evidence will be evaluated using the Grading of Recommendations, Assessment, Development and Evaluation-Confidence in the Evidence from Reviews of Qualitative Research approach. A narrative synthesis of the findings from all included studies will be reported.

**Ethics and dissemination** Ethical approval is not required for this systematic review of existing publicly available literature. We will subsequently aim to publish the results of this systematic review in a peer-reviewed journal.

## STRENGTHS AND LIMITATIONS OF THIS STUDY

⇒ The study's adherence to comprehensive methodology, following established guidelines like the Centre of Research and Dissemination, Meta-analyses of Observational Studies in Epidemiology and Preferred Reporting Items for Systematic Reviews and Meta-Analyses, ensures a systematic and rigorous approach, enhancing credibility and reliability.

⇒ The inclusion of diverse research types (qualitative, quantitative, mixed methods) provides a comprehensive understanding, allowing triangulation of evidence and strengthening the validity of conclusions.

⇒ Independent data management and extraction by two pairs of reviewers reduce biases and enhance objectivity, leading to more robust and credible research outcomes.

⇒ Being a retrospective, observational research design, systematic reviews are subject to systematic and random error. We hope to minimise this by strict adherence to the guidelines mentioned above.

⇒ Potential language bias could lead to oversight of relevant studies, and addressing these limitations is essential for a more comprehensive analysis.

institutions, including medical schools across the UK seems to have fallen dramatically. Teachers and leads across all subjects report a very low attendance rate at small group and larger group teaching and whole cohort lectures and a similar phenomenon has been reported across other medical schools in the UK. Although the literature is still limited on this development, this appears to be a worldwide observation.[1] At our institution and within the literature, this change in the attendance was noted to occur prior to the pandemic,[2] but became significantly more noticeable following the return to in-person teaching postpandemic.

Before the COVID-19 pandemic, most students would have expected to be on clinical placements all day, but the pandemic introduced the need for replacement of in-person education with creative experiments involving

## INTRODUCTION
### Rationale

Postpandemic, following a shift back to in-person teaching, the level of medical student attendance at many higher education

learning online and from home, and the application of telemedicine.[3] This need acted as a catalyst and accelerated the development of online learning. With the shift back to in-person teaching, some students are now actively opting out of in-person teaching.[4] While we acknowledge that the COVID-19 pandemic may have been a factor contributing to this observation, as this was noted prior to the return to in-person teaching, there may be other explanations and the changes in attendance may reflect a shift in the direction of medical education, raising the question of whether attendance is a good measure of high-quality medical education.[5] For example, this could be explained by students increasingly realising that attending core curriculum teaching in person is no longer an absolute necessity for successful completion of the curriculum[1] or an increase in the utility and availability of online learning materials. A greater understanding of what these changes in attendance are, along with why the changes in attendance are happening and its impact on student educational outcomes would be valuable in determining whether further action is required to maintain high standards of medical education.

Identifying areas where institutions can help support this evolving learning process is essential to continue to provide a holistic education. Without this, students' learning may be at risk of becoming overly focused in some areas and less in others. Higher education online learning was available before the COVID-19 pandemic but this was often variable and delivered asynchronously.[6] Exploring previously adopted teaching strategies and their impact on student educational outcomes can help to guide the development of innovative models of teaching to ensure continuous high-quality education.

To date, there are no systematic reviews published that describe the changes in attendance documented within the literature, and its subsequent influence on student education outcomes. This systematic review aims to explore and explain the ongoing changes in medical student attendance that have been noted to start prior to but appear to be exacerbated by the pandemic and describe the potential impacts this may have on student educational outcomes.

### Research question
Research question: in what ways is student attendance to in-person teaching sessions changing and what are the impacts of this on student educational outcomes?

### Objectives
#### Primary aim
► To describe the changes in medical student attendance at in-person teaching sessions
An initial scope of the literature revealed that changes in medical school attendance began prior to the COVID-19 pandemic. As a result, we have decided to remove the secondary aim 'to describe the impact of the COVID-19 pandemic on medical student attendance for in-person teaching sessions' to ensure that

we captured all changes in student attendance, not just those following the pandemic. This leads to the following secondary aims for this literature review.

#### Secondary aims
► To explain the changes to medical student attendance at in-person teaching sessions.
► To describe the impact of the changes in attendance on medical student educational outcomes.
► To describe strategies adopted by educators in light of these changes to medical student attendance at in-person teaching sessions.

### METHODS
The design and methods used for this systematic review comply with Centre of Research and Dissemination Guidelines, Meta-analyses of Observational Studies in Epidemiology (MOOSE) and are reported in line with Preferred Reporting Items for Systematic Reviews and Meta-Analyses (PRISMA). Eligibility criteria were informed using the SPIDER and MOOSE guidelines. This protocol adheres to the PRISMA-Protocol guidelines ('online supplemental material 1.pdf').

### Eligibility criteria
(S) Sample: Medical students in both graduate and undergraduate medical curriculums across all types of universities (private and public).

(PI) The phenomenon of Interest: Changes in attendance, educational outcomes or both.

(E) Evaluation: (1) Recorded or anecdotal evidence of changes in attendance and (2) Comparison between two or more sets of examination scores or any other performance-based measures (including interest, satisfaction and confidence rates).

(D) Design: Primary studies excluding grey literature.

(R) Research type: Primary studies of qualitative, quantitative and mixed-methods research could be searched for, not including systematic reviews, literature reviews or metanalysis.

### Information sources
The search will employ topic-based strategies designed for each database from inception to 20 September 2023. There will be no language or geographical restrictions.

### Databases
EMBASE via OVID, MEDLINE via OVID, Scopus, Web of Science Core Collection, British Education Index via EBSCOhost, ERIC via EBSCOhost.

### SEARCH STRATEGY
The detailed search strategies for the respective databases can be found in online supplemental material 2.pdf.

## STUDY RECORDS
### Data management
Records will be managed through EndNote and NVivo.

### Selection process
#### Inclusion criteria
► Medical students studying in medical schools accredited by their country's governing body.
► Language of publication: all languages.
► Settings: hospitals, medical schools.
► At least one of the following outcomes: (1) attendance rates, (2) performance measures and (3) solutions.

#### Exclusion criteria
► Grey literature.
► Secondary research.

The screening will be performed independently by two pairs of reviewers (SS/SA; SB/AM), with each pair covering 50% of the papers, with a third individual to resolve any disagreements.

### Data collection process
Using a standardised form, two pairs of reviewers (SS/SA; SB/AM) will extract the data independently, with each pair covering 50% of the papers. A third reviewer (PGN) will independently check the data for consistency and clarity.

### Data items
The data extracted will include the following summary data:
► Sample characteristics.
► Sample size.
► Study date.
► Medium of lecture/programme/teaching.
► Stage of medical education training.
► Academic outcome/performance scores.
► Academic outcome/performance measure used.
► Confidence, interest and satisfaction rates.
► Attendance rates (stratified by in-person and otherwise).
► Access rates to online materials.
► Solutions/interventions used.
► Association between attendance rates and academic outcomes/performance.
► Students' perception.
► Other outcomes.

This will be in addition to the critical arguments raised during the process. Both qualitative and quantitative data will be collected for narrative synthesis.

### Outcomes and prioritisation
1. Recorded or anecdotal evidence of changes in attendance at live learning opportunities both in person and otherwise. This would include formative and summative assessments as well if it is used as a 'technique' for learning rather than only as a measurement of performance. An example of an in-person teaching assessment is an OSCE (Objective Structured Clinical Examination) set-up.
2. Comparison of examination scores, performance-based measures or educational outcomes with the implementation of a learning technique or intervention.

### Risk of bias in individual studies
The risk of bias for each included trial will be independently assessed by the same initial reviewers. The fifth reviewer will mediate in situations of disagreement. Cohen's kappa will be used to assess agreement between reviewers. All tools and processes will be piloted before use. The risk of bias will be assessed using the Newcastle-Ottawa Scale.[7 8]

The following assessments will then be made, depending on the study type:

Qualitative studies: The reviewers will assess the appropriateness of data sources and analytical processes, the study's transparency, consideration of context/setting and the study's consideration of researchers' reflexivity and positioning.

Descriptive cross-sectional quantitative studies: The reviewers will assess the appropriateness of the study's sampling strategy and the representativeness of the sample, the use of appropriate measurement instruments and the acceptability of the response rate.

Non-randomised quantitative studies: The reviewers will assess the study's minimisation of selection bias, use of appropriate measurement instruments, use of comparable groups across study conditions and the completeness of outcome data.

### Data synthesis
Qualitative data: Qualitative data will be imported into NVivo software. Malterud's systematic text condensation as thematic analysis will be conducted by two reviewers independently with a discussion following this.[9–12]

Quantitative data: The data collected will undergo narrative synthesis. A meta-analysis will be considered should the quantitative data be sufficiently homogeneous.

### Meta-bias(es)
N/A.

### Confidence in cumulative evidence
Confidence in discrete review findings will be assessed using the recently developed Grading of Recommendations, Assessment, Development and Evaluation-Confidence in the Evidence from Reviews of Qualitative Research.[13] Assessment of confidence in a given review finding involves evaluating how likely it is that the finding represents a real phenomenon, that is, factors leading to changes in attendance, performance and their relationship with each other. This assessment will be based on an evaluation of the following: (1) methodological limitations of the primary studies contributing to the finding, (2) the relevance of the primary contributing studies regarding the objectives of the systematic review, (3) the

**Table 1** Estimated timeline

| Stage | Planned period |
|---|---|
| Title and abstract screening | December 2023 |
| Full-text screening | December 2023 |
| Data extraction and quality assessment | January 2024 |
| Statistical analysis | February 2024 |
| Manuscript writing, revision and submission | September 2024 |

coherence of the finding and (4) the adequacy of data supporting the finding.

A summary table will list each review finding—primary contributing studies, evaluations of the above four domains, an overall confidence rating (high, moderate, low or very low) and an explanation of the rating judgement.

## Ethics and dissemination

Ethical approval is not required for this systematic review of existing publicly available literature. We will subsequently aim to publish the results of this systematic review in a peer-reviewed journal.

## Estimated timeline

The estimated timeline of the systematic review is as described in table 1.

**Contributors** PGN: drafted the proposal and conceptualisation of topic. SB: data collection and refining of topic. AM: data collection and refining of topic. SS: data collection and refining of topic. SA: data collection and refining of topic. SRXT: conceptualisation of topic. IK: review and critiquing to improve research methodology with the drafting of the search strategy. CT: conceptualisation of topic and supervision. EL: conceptualisation of topic and supervision.

**Funding** The authors have not declared a specific grant for this research from any funding agency in the public, commercial or not-for-profit sectors.

**Competing interests** None declared.

**Patient and public involvement** Patients and/or the public were not involved in the design, or conduct, or reporting, or dissemination plans of this research.

**Patient consent for publication** Not applicable.

**Provenance and peer review** Not commissioned; externally peer reviewed.

**ORCID iDs**
Palaniappan Ganesh Nagappan http://orcid.org/0000-0002-9203-8836
Samuel Brown http://orcid.org/0009-0000-8484-5411
Alex McManus http://orcid.org/0009-0007-1398-4212
Sarah Sayers http://orcid.org/0009-0003-8931-7361
Shazia Absar http://orcid.org/0009-0008-8803-0496
Sapphire Rou Xi Tan http://orcid.org/0000-0002-1462-3263
Edward Lau http://orcid.org/0000-0002-8595-2734
Charlotte Tulinius http://orcid.org/0000-0002-1322-4703

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
