## [Reviewer comments · BMJ Open]

ARTICLE DETAILS

TITLE (PROVISIONAL)	Changes in medical student attendance and its impact on student educational outcomes – A systematic review protocol
AUTHORS	Nagappan, Palaniappan Ganesh; Brown, Samuel; McManus, Alex; Sayers, Sarah; Absar, Shazia; Tan, Sapphire Rou Xi; Kuhn, Isla; Tulinius, Charlotte; Lau, Edward

VERSION 1 – REVIEW

REVIEWER	May Ohn University Malaysia Sabah, medicine
REVIEW RETURNED	20-Aug-2023

GENERAL COMMENTS	Study objective is to investigate why attendance is decreased in post-pandemic period while increasing online learning. It's contradict to study strategy of database from inception. Keyword should include: medical students as well as doctors? In inclusion criteria, medical students only or include postgraduate as medical trainees are postgraduate. Ethic statement isn't clear. 2 table search results of 159 and 76 are pretty huge different number from above 4 database result numbers despite similar search strategy. Is it possible present all in tables? If consort diagram is inserted, it would be more understandable on number of eligible and included studies extracted from each database. (GRADE-CERQual) approach by Lewin et al reference is missing.
---

REVIEWER	Rebecca Donkin University of the Sunshine Coast, School of Health and Sport Sciences
REVIEW RETURNED	22-Aug-2023

GENERAL COMMENTS	The topic of the protocol is of interest however there is not a clear research question which confounds the approach to the method. Objectives are provided however, I would suggest providing clear research questions that can be answered through a review. Interestingly, the authors have provided the PRISMA-P statement "Provide an explicit statement of the question(s) the review will address" but have not completed this step. The authors suggest a systematic review but I am concerned that the number of articles available post-pandemic to assess this will not provide the depth and breadth for a systematic review. I am also curious why the database search is from inception to 1st Feb 2023 given the introduction includes the aim to understand the reasons behind the online shift post-pandemic, wouldn't it be appropriate to review papers post-pandemic?
---

	The Introduction requires more referencing and description of the gap or problem. The following sentence with only 1 reference is not appropriate "During the pandemic, the downfalls of online teaching were seen as devastating for both students and teachers (2), and it seems almost paradoxical that students are now actively opting out of in-person teaching". Under Outcomes it is unclear what "attendance" means. Is this compulsory attendance, assessment attendance, lecture attendance, formative or summative assessment attendance? For "Comparison of examination scores or any other performance-based measures", is this pre and post pandemic or attended/non-attended or in-person/online? It is unclear if the following statement is part of the protocol or a subsequent research study that would require ethics "The collated and analysed data will be published and subsequently be used to guide focused group discussion".
--	---

REVIEWER	SRIKANTH U The University of the West Indies
REVIEW RETURNED	29-Aug-2023

GENERAL COMMENTS	This provides a valuable protocol for teaching in medical stream.
---

VERSION 1 – AUTHOR RESPONSE

Reviewer 1	
Study objective is to investigate why attendance is decreased in post-pandemic period while increasing online learning. It's contradict to study strategy of database from inception. Keyword should include: medical students as well as doctors? In inclusion criteria, medical students only or include postgraduate as medical trainees are postgraduate.	The objectives have been clarified in the manuscript. Our scope is looking at medical education prior to qualifying for MBBS, MD, or equivalent. As such, we have not included doctors within our search criteria. Understandably, some medical students are postgraduate students in several countries, however, the MeSH heading for "Education, Medical, Graduate" refers to individuals that have already received their medical degree which does not fall into our scope. Below is an excerpt from Medline via Ovid. For the MeSH HEADING: EDUCATION, MEDICAL, GRADUATE SCOPE: "Educational programs for medical graduates entering a specialty. They include formal

	specialty training as well as academic work in the clinical and basic medical sciences and may lead to board certification or an advanced medical degree.”
Ethic statement isn't clear.	Thank you, we have now amended this to make it clearer as ethical approval is not required for this systematic review of existing available literature.
2 table search results of 159 and 76 are pretty huge different number from above 4 database result numbers despite similar search strategy. Is it possible present all in tables?	Yes, it is possible to present all data in tables, we have amended the manuscript to reflect this. The differences in results could be attributed to ERIC and BEI having significantly different search coverages as databases, with ERIC and BEI primarily focused on education-related papers; Medline, Embase and Web of Science are instead focused on biomedical sciences and related scientific fields.
If consort diagram is inserted, it would be more understandable on number of eligible and included studies extracted from each database.	This will be included in the manuscript for the completed systematic review as we are still at the protocol drafting stage
(GRADE-CERQual) approach by Lewin et al reference is missing.	The appropriate reference has been added.
Reviewer 2	
The topic of the protocol is of interest however there is not a clear research question which confounds the approach to the method. Objectives are provided however, I would suggest providing clear research questions that can be answered through a review. Interestingly, the authors have provided the PRISMA-P statement "Provide an explicit statement of the question(s) the review will address" but have not completed this step.	The primary and secondary aims have now been described alongside the research question this systematic review hopes to answer.
The authors suggest a systematic review but I am concerned that the number of articles available post-pandemic to asses this will not provide the depth and breadth for a systematic review. I am also curious why the database search is from inception to 1st Feb 2023 given the introduction includes the aim to understand the reasons behind the online shift post-pandemic, wouldn't it be	We have updated our introduction and research question to more clearly reflect this. Our focus is on how attendance and/or performance have changed with various interventions over time. The pandemic may have acted as

appropriate to review papers post-pandemic?	a catalyst for change, but this is unclear until the data has been reviewed. By looking at the effects on attendance rates and performance measures with various interventions over time with the pandemic as a key time point, we could interpret the effect the pandemic (as a catalyst) has had on attendance rates and performance measures. From here, we can also look at what has worked before, what has worked more recently and what no longer works anymore.
The Introduction requires more referencing and description of the gap or problem. The following sentence with only 1 reference is not appropriate "During the pandemic, the downfalls of online teaching were seen as devastating for both students and teachers (2), and it seems almost paradoxical that students are now actively opting out of in-person teaching".	Thank you for the recommendation, the manuscript has now been reflected to make it clearer.
Under Outcomes it is unclear what "attendance" means. Is this compulsory attendance, assessment attendance, lecture attendance, formative or summative assessment attendance? For "Comparison of examination scores or any other performance-based measures", is this pre and post pandemic or attended/non-attended or in-person/online?	The manuscript has now been amended to make it clearer. It includes placements, lectures, and seminars. It would include formative and summative assessments as well if it is used as a "technique" for learning rather than only as a final measurement of performance. This refers to both pre- and post-pandemic. Both in-person and online attendance would be recorded for analysis discreetly. The focus would be following the primary aim for in-person teaching which has now been clarified in the manuscript.
It is unclear if the following statement is part of the protocol or a subsequent research study that would require ethics "The collated and analysed data will be published and subsequently be used to guide focused group discussion".	Thank you, we have now amended this to make it clearer as ethical approval is not required for this systematic review of existing available literature.

Reviewer 3	
This provides a valuable protocol for teaching in medical stream.	Thank you

VERSION 2 – REVIEW

REVIEWER	Rebecca Donkin University of the Sunshine Coast, School of Health and Sport Sciences
REVIEW RETURNED	17-Oct-2023

GENERAL COMMENTS	Thank you for the opportunity to review the manuscript "Changes in medical student attendance and its impact on student educational outcomes – A systematic review protocol". The review is well written however, the major flaw is the research question and the reasoning behind starting the search "from inception" for a post pandemic question. I have added some feedback/comments to assist in improving the paper.  1. Research question is awkwardly worded. Could it be revised to something such as "How has medical student attendance to in-person teaching sessions changed post COVID-19 pandemic and what are the impacts on student educational outcomes?" 2. As mentioned by a previous reviewer (1) it seems inappropriate to include the study strategy of a database from inception when the investigation is why attendance is decreased in the post-pandemic period. I recommend updating this date to include post-pandemic studies of non/poor attendance when return to in-person. Otherwise the papers retrieved could be misleading. 3. Again Reviewer 2 from a previous report has stated the above "why is the database search from inception". The explanation provided by the authors is not adequate and does not align to the research question. I am assuming that there are not enough papers post-pandemic to include in a systematic review so the data has been expanded pre Covid but then this does not agree with the research question.
--

VERSION 2 – AUTHOR RESPONSE

Reviewer 2	
Research question is awkwardly worded. Could it be revised to something such as "How has medical student attendance to in-person teaching sessions changed post COVID-19 pandemic and what are the impacts on student educational outcomes?" As mentioned by a previous reviewer (1) it seems inappropriate to include the study strategy of a database from inception when the investigation is why attendance is decreased in the post-pandemic period. I recommend updating this date to include post-pandemic studies of non/poor attendance when return to in-person. Otherwise the papers retrieved could be misleading. Again Reviewer 2 from a previous report has stated the above "why is the database search from inception". The explanation	Dear Dr Donkin, Thank you for your time and valuable comments. We will address all three points in one go, as they are closely linked. Changes in the level of medical student attendance has been noted within the literature for decades, not just post pandemic, which we have now explained in the introduction and referenced. We have also noted these changes within our own institution and educational practice. However, following the return of in-person teaching post COVID-

provided by the authors is not adequate and does not align to the research question. I am assuming that there are not enough papers post-pandemic to include in a systematic review so the data has been expanded pre Covid but then this does not agree with the research question.

19 pandemic, it appears that the changes have been exacerbated and more noticeable. The systematic review will therefore describe the changes in medical student attendance over time, not just after the COVID-19 pandemic, and therefore will not include COVID-19 in its wording. In order to capture all relevant papers in the search, including papers that describe changes in attendance prior to COVID-19, the search is from database inception as oppose to post pandemic. We believe this answers questions 1, 2 and 3 raised by all previous reviewers.